# Algorithmic Recourse in Abnormal Multivariate Time Series

**Xiao Han**                                                                                              *xiao.han@usu.edu*
*Utah State University*

**Lu Zhang**                                                                                               *lz006@uark.edu*
*University of Arkansas*

**Yongkai Wu**                                                                                        *yongkaw@clemson.edu*
*Clemson University*

**Shuhan Yuan**                                                                                      *shuhan.yuan@usu.edu*
*Utah State University*

**Reviewed on OpenReview:** *https://openreview.net/forum?id=kzxFc2Suo5*

## Abstract

Algorithmic recourse provides actionable recommendations to alter unfavorable predictions of machine learning models, enhancing transparency through counterfactual explanations. While significant progress has been made in algorithmic recourse for static data, such as tabular and image data, limited research explores recourse for multivariate time series, particularly for reversing abnormal time series. This paper introduces Recourse in time series Anomaly Detection (RecAD), a framework for addressing anomalies in multivariate time series using backtracking counterfactual reasoning. By modeling the causes of anomalies as external interventions on exogenous variables, RecAD predicts recourse actions to restore normal status as counterfactual explanations, where the recourse function, responsible for generating actions based on observed data, is trained using an end-to-end approach. Experiments on synthetic and real-world datasets demonstrate its effectiveness.

## 1 Introduction

Algorithmic recourse refers to the process of offering individuals or entities actionable recommendations to change undesirable outcomes generated by predictive models (Karimi et al., 2022). In the context of counterfactual explanations, algorithmic recourse aims to identify minimal and feasible modifications to input features that can shift a model's prediction (Verma et al., 2024), which enhances transparency and interpretability in machine learning systems.

In recent years, algorithmic recourse has been extensively studied in the context of tabular (Karimi et al., 2020; Chen et al., 2020; Creager & Zemel, 2023; Gao & Lakkaraju, 2023; De Toni et al., 2024) and image data (Jung et al., 2022; Wang & Vasconcelos, 2020; Von Kügelgen et al., 2023). Despite significant advancements in these areas, there is a noticeable lack of research on algorithmic recourse for time series. Time series data, which consist of sequential observations over time, are ubiquitous in domains such as industrial systems (sensor readings), finance (stock prices), and healthcare (patient vital signs). The temporal dependencies and complex dynamics inherent in time series data present unique challenges that differ from those in static tabular or image data, as interventions made at one point in time can lead to cascading effects across subsequent periods, making it essential to accurately account for temporal and causal dependencies when designing recourse actions.

This paper studies algorithmic recourse for time series data. While our method could serve as a general framework, in this paper, we focus on abnormal multivariate time series due to their practical significance. Abnormal time series are characterized by anomalies or unexpected patterns across multiple variables. Such

anomalies can indicate critical events such as system failures, fraudulent activities, or health deterioration. Providing algorithmic recourse in this context means offering explanations and actionable strategies to mitigate or prevent undesirable outcomes reflected by the anomalies, advancing anomaly detection beyond mere identification toward practical, intervention-driven solutions. For example, Figure 1 presents the usage data of two control nodes, nodes 117 and 124, in an OpenStack testbed (Nedelkoski et al., 2020). Each node's performance metrics include CPU and memory usage. When an anomaly detection model triggers an anomaly alert (indicated by the red area in the top figure), a recourse action is recommended to free up memory usage on node 124, aiming to correct the abnormal behavior. After implementing the recourse action (shown as the green area in the bottom figure), the system returns to normal, as indicated by the dashed lines in the bottom figure. This example demonstrates how algorithmic recourse can provide quick and cost-effective remediation of the issue when an anomaly detection model detects an abnormal status.

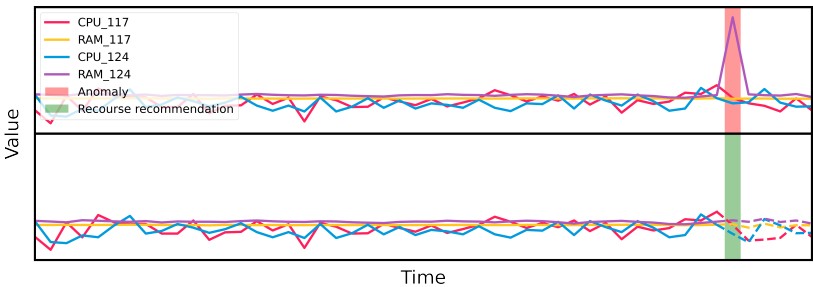

Figure 1: Recourse recommendations for flipping an abnormal status of a distribution system.

To recommend recourse in abnormal time series, we treat anomalies as external interventions on exogenous variables in a structural causal model. For instance, in a distributed computing environment, a sudden surge in incoming traffic to a particular node, such as caused by an external request spike, may lead to abnormal CPU or memory usage. Such anomaly arises not from a change in the system's internal dynamics but from an external factor influencing the exogenous inputs to the node. Methodologically, we align this formulation of anomalies with the concept of backtracking counterfactual reasoning (Von Kügelgen et al., 2023), which we adopt as the foundation of our method. Backtracking counterfactuals involve reasoning backward from an observed outcome to infer the necessary changes in exogenous variables that could have led to a different result, assuming that all causal relationships remain intact. This approach is particularly suitable for interpreting abnormal states in time series, as it allows anomalies to be traced back systematically to plausible external interventions, providing explanatory insights and actionable recourse strategies.

Specifically, we propose a neural network-based framework for algorithmic **Rec**ourse in time series **A**nomaly **D**etection (RecAD). Our method assumes that the causal relationships within the time series have been modeled using Granger causality discovery methods, such as Generalized Vector Autoregression (GVAR) (Marcinkevičs & Vogt, 2021). Given a time series with abnormal segments, RecAD addresses the question: *What is the most likely external intervention that explains these abnormal states?* Leveraging backtracking counterfactual reasoning, we identify recourse actions that mitigate anomalies by restoring abnormal states to normal ones. We formulate this task as a constrained maximum likelihood problem, where the identified actions serve both as plausible counterfactual explanations and actionable interventions. To enable end-to-end learning of a recourse function, we parameterize the recourse function with neural networks, mapping observed data to recourse actions. We adopt the Abduction-Action-Prediction procedure to model the downstream effects of interventions, deriving the counterfactual posterior through cross-world abduction. A differentiable loss function is then defined to guide the learning of the recourse function. Furthermore, we discuss practical considerations, such as handling sequence-level anomalies and addressing anomalies caused by interventions on structural equations rather than on exogenous variables. Empirical studies on two synthetic and one real-world datasets demonstrate the effectiveness of RecAD in recommending recourse actions for restoring abnormal time series.

The contribution of this paper can be summarized as follows. 1) We propose Recourse in time series Anomaly Detection (RecAD), a novel framework that generates recourse actions to correct anomalies in multivariate time series data; 2) by treating anomalies due to external interventions on exogenous variables, we apply

the concept of backtracking counterfactuals to time series analysis and formulate algorithmic recourse as a constrained maximum likelihood problem using backtracking counterfactual reasoning; and 3) we addressing practical considerations and demonstrating effectiveness through empirical studies using both synthetic and real-world datasets.

## 2    Related Work

A time series anomaly is defined as a sequence of data points that deviates from frequent patterns in the time series (Schmidl et al., 2022). Recently, a large number of approaches have been developed for time series anomaly detection (Schmidl et al., 2022; Blázquez-García et al., 2021). However, explaining the detection results is under-exploited (Jacob et al., 2020; Kieu et al., 2022). Algorithmic recourse can provide counterfactual explanations by recommending actions to reverse unfavorable outcomes from an automated decision-making system (Karimi et al., 2022). Specifically, given a predictive model and a sample having an unfavorable prediction from the model, algorithmic recourse is to identify the minimal consequential recommendation that leads to a favorable prediction from the model. The key challenge of identifying the minimal consequential recommendation is to consider the causal relationships governing the data. Any recommended actions on a sample should be carried out via structural interventions leading to a counterfactual instance. Multiple algorithmic recourse algorithms on binary classification models have been developed (Karimi et al., 2020; 2021; von Kügelgen et al., 2022; Dominguez-Olmedo et al., 2022). Recently, algorithmic recourse for anomaly detection on tabular data has also been discussed (Datta et al., 2022). However, this study does not consider causal relationships when generating counterfactuals.

In this work, we focus on addressing the algorithmic recourse for anomaly detection in multivariate time series, considering causal relationships. Specifically, we assume that anomalies result from external interventions, distinguishing our approach from typical counterfactual reasoning. We align our approach with backtracking counterfactuals (Von Kügelgen et al., 2023), which suggests a backtracking interpretation of counterfactuals where causal laws remain unchanged in the counterfactual world, and differences from the factual world are attributed to changes in exogenous variables that alter the initial conditions. Different from a recent study (Kladny et al., 2024), which also leverages backtracking counterfactual reasoning to generate counterfactual explanations for static high-dimensional data by developing a deep generative model, our work focuses on providing backtracking counterfactual explanations in abnormal multivariate time series.

## 3    Preliminary

**Granger Causality**. Granger causality (Granger, 1969; Dahlhaus & Eichler, 2003) is a standard approach for capturing causal relationships in multivariate time series. Formally, let a stationary time-series be $\mathcal{X} = (\mathbf{x}_1, \ldots, \mathbf{x}_t, \ldots, \mathbf{x}_T)$, where $\mathbf{x}_t \in \mathbb{R}^d$ is a $d$-dimensional vector (e.g., $d$-dimensional time series from $d$ sensors) at a specific time $t$. Define the true data generation mechanism in the form of

$$x_t^{(j)} := f^{(j)}(\mathbf{x}_{\leq t-1}^{(1)}, \cdots, \mathbf{x}_{\leq t-1}^{(d)}) + u_t^{(j)}, \text{ for } 1 \leq j \leq d, \tag{1}$$

where $\mathbf{x}_{\leq t-1}^{(j)} = [\cdots, x_{t-2}^{(j)}, x_{t-1}^{(j)}]$ denotes the present and past of series $j$; $u_t^{(j)}$ indicates exogenous variable of time series $j$ at time step $t$; $\mathcal{F} = \{f^{(1)}, \ldots, f^{(d)}\}$ is a set of unknown functions, and $f^{(j)}(\cdot) \in \mathcal{F}$ is the function for time series $j$ that captures how the past values impact the future values of $\mathbf{x}^{(j)}$. Then, the time series $i$ Granger causes $j$, if $f^{(j)}$ depends on $\mathbf{x}_{\leq t-1}^{(i)}$, i.e., $\exists \mathbf{x'}_{\leq t-1}^{(i)} \neq \mathbf{x}_{\leq t-1}^{(i)} : f^{(j)}(\mathbf{x}_{\leq t-1}^{(1)}, \cdots, \mathbf{x'}_{\leq t-1}^{(i)}, \cdots, \mathbf{x}_{\leq t-1}^{(d)}) \neq f^{(j)}(\mathbf{x}_{\leq t-1}^{(1)}, \cdots, \mathbf{x}_{\leq t-1}^{(i)}, \cdots, \mathbf{x}_{\leq t-1}^{(d)})$.

Granger causal discovery, i.e., learning Granger causal relationships from the observational data, has been extensively studied (Nauta et al., 2019; Tank et al., 2021; Marcinkevičs & Vogt, 2021). While multiple methods are available, in this paper, we utilize a generalized vector autoregression (GVAR) approach that can model nonlinear Granger causality (Marcinkevičs & Vogt, 2021). GVAR models the Granger causality of the $t$-th time step given the past $K$ lags by

$$\mathbf{x}_t = \sum_{k=1}^{K} g_k(\mathbf{x}_{t-k})\mathbf{x}_{t-k} + \mathbf{u}_t, \tag{2}$$

where $g_k(\cdot) : \mathbb{R}^d \to \mathbb{R}^{d \times d}$ is a feedforward neural network predicting a coefficient matrix at time step $t - k$; $\mathbf{u}_t$ is the exogenous variable for time step $t$. The element $(i, j)$ of the coefficient matrix from $g_k(\mathbf{x}_{t-k})$ indicates the influence of $x_{t-k}^{(j)}$ on $x_t^{(i)}$. Meanwhile, $K$ neural networks are used to predict $\mathbf{x}_t$. Therefore, relationships between $d$ variables over $K$ time lags can be explored by inspecting $K$ coefficient matrices. The $K$ neural networks are trained by the objective function:

$$\mathcal{L} = \frac{1}{T - K} \sum_{t=K+1}^{T} \|\mathbf{x}_t - \hat{\mathbf{x}}_t\|_2 + \frac{\lambda}{T - K} \sum_{t=K+1}^{T} R(\mathcal{M}_t) + \frac{\gamma}{T - K - 1} \sum_{t=K+1}^{T-1} \|\mathcal{M}_{t+1} - \mathcal{M}_t\|_2,$$

where $\hat{\mathbf{x}}_t = \sum_{k=1}^{K} g_k(\mathbf{x}_{t-k})\mathbf{x}_{t-k}$ indicates the predicted value by GVAR; $\mathcal{M}_t := [g_K(\mathbf{x}_{t-K}) : g_{K-1}(\mathbf{x}_{t-K+1}) : \cdots : g_1(\mathbf{x}_{t-1})]$ indicates the concatenation of generalized coefficient matrices over the past the $K$ time steps; $R(\cdot)$ is the penalty term for sparsity, such as L1 or L2 norm; the third term is a smoothness penalty; $\lambda$ and $\gamma$ are hyperparameters. After training, the generalized coefficient predicted by $g_k(\mathbf{x}_{t-k})$ indicates the causal relationships between time series at the time step $t - k$.

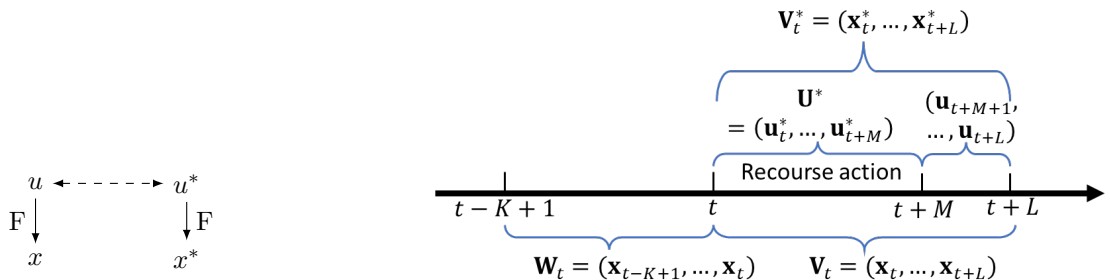

(a) Backtracking counterfactual.      (b) Algorithmic recourse in multivariate time series.

Figure 2: Illustration of backtracking counterfactual and algorithmic recourse.

**Counterfactual Reasoning**. Counterfactual inference provides a framework for understanding causal relationships by exploring counterfactual scenarios, often framed as "what-if" questions. For example, "would the CPU utilization stay within safe operating limits if the memory usage followed its default profile?" In causal inference, counterfactuals are typically analyzed using two main approaches: interventional counterfactuals (Pearl, 2009) and backtracking counterfactuals (Von Kügelgen et al., 2023). In interventional counterfactuals, causal laws are modified while the factual and counterfactual worlds share the same background conditions. In backtracking counterfactuals, on the other hand, as shown in Figure 2a, causal laws remain consistent, but the counterfactual explanation adjusts exogenous variables to account for changes observed in the outcome. Both types of counterfactual reasoning are valuable for understanding causal mechanisms; however, backtracking counterfactuals are especially useful for explaining observed anomalies by inferring the most plausible prior conditions that led to the current state. This insight forms the basis of our method.

## 4  Methodology

In this section, we propose the algorithmic **Rec**ourse in time series **A**nomaly **D**etection (RecAD) method to compute algorithmic recourse actions for anomalies in multivariate time series, assuming that the causal relationships have been captured using a Generalized Vector Autoregression (GVAR) model as shown in Eq. (2). We first provide the general framework of the problem formulation based on the backtracking counterfactual. Then, we explain the detailed implementation of each component of the framework.

**External intervention vs. structural intervention.** Depending on how anomalies occur within a causal mechanism, they can be divided into two categories: those due to changes in exogenous variables (external interventions) and those due to changes in the structural equations (structural interventions). In this paper, we focus on anomalies caused by external interventions and briefly discuss in Section 4.5.2 how to handle anomalies resulting from structural interventions.

### 4.1 Problem Formulation: Algorithmic Recourse based on Backtracking Counterfactual

Following the common setting for time series anomaly detection (Audibert et al., 2020; Tuli et al., 2022), given a multivariate time series $\mathcal{X}$, we consider a local window with length $K$ as $\mathbf{W}_t = (\mathbf{x}_{t-K+1}, ..., \mathbf{x}_t)$ and convert a time series $\mathcal{X}$ to a sequence of sliding windows $\mathcal{W} = (\mathbf{W}_K, \mathbf{W}_{K+1}, ..., \mathbf{W}_T)$. Consider a score-based anomaly detection function $s(\cdot)$ that takes a time window $\mathbf{W}_t$ as the input. If $s(\mathbf{W}_t) > \tau$, then the time step $t$ will be labeled as in an abnormal state. Assume that an anomaly occurs at time $t$, which causes abnormal states in a time window $\mathbf{V}_{t+L} = (\mathbf{x}_t, ..., \mathbf{x}_{t+L})$ (assume that $L < K$) due to the ripple effect. By treating the anomaly as an external intervention on the exogenous variable $\mathbf{u}_t$, our objective is to find the recourse action $\boldsymbol{\theta}_t$ at the time step $t$ to reverse the abnormal states in the time window $\mathbf{V}_{t+L}$ via counterfactual reasoning. Such action can be viewed as the most plausible counterfactual explanation for the abnormal states. To simplify our discussions, we mainly focus on point anomalies. Extending our approach to sequence anomalies is straightforward and will be addressed in Section 4.5.1.

Specifically, the backtracking counterfactual computes the probability of the exogenous variables in a counterfactual world, which may differ from their factual counterparts, given both the factual and counterfactual endogenous variables. As shown in Figure 2b, in our context, the factual endogenous variables are $\mathbf{W}_t$, i.e., the observational data. We denote the backtracking counterfactual exogenous variable at time $t$ by $\mathbf{u}_t^*$, which is obtained by performing the recourse action, i.e., $\mathbf{u}_t^* = \mathbf{u}_t + \boldsymbol{\theta}_t$, where $\mathbf{u}_t$ is the factual exogenous variable. We then denote the counterfactual endogenous variables by $\mathbf{V}_{t+L}^* = (\mathbf{x}_t^*, ..., \mathbf{x}_{t+L}^*)$, i.e., the counterfactual variants of $\mathbf{V}_{t+L}$ obtained by performing the recourse action, with the requirement that their states are restored to normal. As a result, the problem of algorithmic recourse can be formulated to maximize the likelihood of $\mathbf{u}_t^*$ conditioning on $\mathbf{W}_t$, subject to the constraints that $s(\mathbf{V}_{t'}^*) < \tau$ for $t' \in [t, t+L]$.

**Problem Formulation 1** *Given local windows of time series* $\mathbf{W}_t = (\mathbf{x}_{t-K+1}, ..., \mathbf{x}_t)$ *and a score function* $s(\cdot)$ *with time step $t$ labeled as abnormal, i.e.,* $s(\mathbf{W}_t) > \tau$, *the algorithmic recourse aims to find the recourse action at time $t$, i.e.,* $\mathbf{u}_t^* = \mathbf{u}_t + \boldsymbol{\theta}_t$, *to restore normal states in a future time window* $\mathbf{V}_{t+L}^* = (\mathbf{x}_t^*, ..., \mathbf{x}_{t+L}^*)$, *by solving the following constrained maximization likelihood problem:*

$$\arg\max_{\boldsymbol{\theta}_t} P(\mathbf{u}_t^* \mid \mathbf{W}_t) \quad s.t. \quad s(\mathbf{V}_{t'}^*) < \tau, \quad t' \in [t, t+L].$$

By introducing penalty terms to the negative log-likelihood to enforce constraints, we derive the revised problem formulation as follows.

**Problem Formulation 2** *The algorithmic recourse aims to solve the optimization problem:*

$$\arg\min_{\boldsymbol{\theta}_t} \mathcal{L}(\boldsymbol{\theta}_t) = \sum_{t'=t}^{t+L} \max\left\{s(\mathbf{V}_{t'}^*) - \tau, 0\right\} - \lambda \log P(\mathbf{u}_t^* \mid \mathbf{W}_t). \tag{3}$$

Next, we discuss the implementation of each component of the problem formulation to enable an end-to-end learning of a recourse function.

### 4.2 Implementing Recourse Function

To generate effective recourse actions, we learn a recourse function $h_\phi(\cdot)$ parameterized by $\phi$ that maps from the observed time series to a perturbation of the exogenous variables, i.e., $\theta_t = h_\phi(W_{t-1}, \Delta_t)$, where $\theta_t$ is the proposed intervention at time step $t$.

The design of $h_\phi$ is guided by two key considerations. 1) **Temporal context** is essential for accurate intervention, as the causal dependencies in multivariate time series often span multiple time lags. Therefore, we encode the preceding time window $\mathbf{W}_{t-1}$ using a Long Short-Term Memory (LSTM) network to capture latent temporal features and long-range dependencies. 2) **Deviation from expected dynamics** is another critical signal for recourse. We define the deviation term $\Delta_t = x_t - \hat{x}_t$, where $\hat{x}_t$ is the expected value at time $t$ based on GVAR. Intuitively, $\Delta_t$ quantifies how much the current state deviates from its normal trajectory

and serves as a proxy for anomalous influence from exogenous variables. Concretely, our implementation is structured as follows:

$$\mathbf{z}_{t-1} = LSTM(\mathbf{W}_{t-1}) \quad \mathbf{z}_{\Delta} = FFNN(\Delta_t) \quad \boldsymbol{\theta}_t = FFNN(\mathbf{z}_{t-1} \oplus \mathbf{z}_{\Delta}), \tag{4}$$

where $\oplus$ denotes vector concatenation. The final feedforward network combines the latent temporal state and the deviation signal to produce the recourse action.

We empirically validate the necessity of each component through ablation studies, showing that removing either the LSTM or the deviation encoder significantly degrades recourse effectiveness.

### 4.3 Inferring Downstream Effect

We then infer the downstream effect $\mathbf{V}_{t'}^*$ because, based on Granger causality, the recourse action at time $t$ leads to the counterfactual variants of the following time steps, i.e., $\mathbf{V}_{t+L}^* = (\mathbf{x}_t^*, ..., \mathbf{x}_{t+L}^*)$. To compute the counterfactual variants, according to the additive noise assumption presented in Eq. (1), given backtracking counterfactual $\mathbf{u}_t^*$, we directly have $\mathbf{x}_t^* = \mathbf{x}_t - \mathbf{u}_t + \mathbf{u}_t^* = \mathbf{x}_t + \boldsymbol{\theta}_t$. However, since we perform the recourse action only at time $t$, it remains necessary to infer the factual exogenous variables for all $t' > t$. Thus, we proceed to utilize the Abduction-Action-Prediction (Pearl, 2009) procedure to compute $\mathbf{x}_{t'}^*$. Formally, based on the causal relationships learned by GVAR shown in Eq. (2), the Abduction-Action-Prediction (AAP) procedure to compute $\mathbf{x}_{t'}^*$ for $t' = t+1, \cdots, t+L$ can be described below.

**Step 1 (Abduction):**

$$\mathbf{u}_{t'} = \mathbf{x}_{t'} - \sum_{k=1}^{K} g_k(\mathbf{x}_{t'-k})\mathbf{x}_{t'-k} \tag{5}$$

**Step 2 (Action): Not required**

**Step 3 (Prediction):**

$$\mathbf{x}_{t'}^* = \sum_{k=1}^{t'-t} g_k(\mathbf{x}_{t'-k}^*)\mathbf{x}_{t'-k}^* + \sum_{k=t'-t+1}^{K} g_k(\mathbf{x}_{t'-k})\mathbf{x}_{t'-k} + \mathbf{u}_{t'}. \tag{6}$$

In essence, the abduction is to derive the exogenous variable $\mathbf{u}_{t'}$ at step $t'$ based on the observed value, then the prediction is to predict the counterfactual value $\mathbf{x}_{t'}^*$ at step $t'$ after conducting action $\boldsymbol{\theta}_t$ at step $t$.

### 4.4 Deriving Counterfactual Posterior

According to Problem Formulation 2, the next step is to derive the counterfactual posterior $P(\mathbf{u}_t^* \mid \mathbf{W}_t)$. Following the backtracking counterfactual (Von Kügelgen et al., 2023), we apply cross-world abduction to obtain:

$$\begin{aligned}
P(\mathbf{u}_t^* \mid \mathbf{W}_t) &= \sum_{\mathbf{u}_t'} P(\mathbf{u}_t^*, \mathbf{u}_t' \mid \mathbf{W}_t) = \sum_{\mathbf{u}_t'} \frac{P(\mathbf{u}_t^*, \mathbf{u}_t') P(\mathbf{W}_t \mid \mathbf{u}_t^*, \mathbf{u}_t')}{P(\mathbf{W}_t)} \\
&\sim \sum_{\mathbf{u}_t'} P(\mathbf{u}_t^*, \mathbf{u}_t') P(\mathbf{W}_t \mid \mathbf{u}_t^*, \mathbf{u}_t) = \sum_{\mathbf{u}_t'} P(\mathbf{u}_t^*, \mathbf{u}_t') P(\mathbf{W}_t \mid \mathbf{u}_t') \\
&= \sum_{\mathbf{u}_t'} P(\mathbf{u}_t^* \mid \mathbf{u}_t') P(\mathbf{W}_t, \mathbf{u}_t') = P(\mathbf{u}_t^* \mid \mathbf{u}_t) I(\mathbf{W}_t(\mathbf{u}_t) = \mathbf{W}_t),
\end{aligned}$$

where the second equality is based on the Bayes rule, the third equality is due to the fact that $\mathbf{u}_t^*$ and $\mathbf{W}_t$ are conditionally independent given $\mathbf{u}_t$ (obtained using d-separation in Figure 2a), $I(\cdot)$ is the indicator function, $\mathbf{u}_t$ is the factual exogenous variable obtained by abduction, and $P(\mathbf{u}_t^* \mid \mathbf{u}_t)$ is the backtracking conditional. Following the suggestion in (Von Kügelgen et al., 2023), we construct $P(\mathbf{u}_t^* \mid \mathbf{u}_t)$ based on the distance between $\mathbf{u}_t^*$ and $\mathbf{u}_t$, defined as $P(\mathbf{u}_t^* \mid \mathbf{u}_t) = 1/Z \cdot \exp\{-d(\mathbf{u}_t^*, \mathbf{u}_t)\}$ where $Z$ is a normalization constant. By using the squared Mahalanobis under isotropic covariance, as a result, we have that $P(\mathbf{u}_t^* \mid \mathbf{W}_t) \sim \exp\{-\|\mathbf{u}_t^* - \mathbf{u}_t\|_2^2\} = \exp\{-\|\boldsymbol{\theta}_t\|_2^2\}$.

Combining all the components above, we obtain our final loss function:

$$\mathcal{L}(\phi) = \sum_{t'=t}^{t+L} \max\left\{s(\mathbf{V}_{t'}^*) - \tau, 0\right\} + \lambda\|\boldsymbol{\theta}_t\|_2^2, \tag{7}$$

where $\phi$ are the parameters of the recourse function defined in Eq. 4 and $\lambda$ is a hyperparameter.

### 4.5 Other Considerations

#### 4.5.1 From Point Anomalies to Sequence Anomalies

So far, we have introduced our framework in the setting point anomalies, where a single abnormal state is explained by one recourse action $\mathbf{u}_t^*$. This allows us to present the core intuition and mechanics of our approach in a straightforward manner. In practice, however, abnormal states in time series data are often caused by a sequence of anomalies, which requires extending our formulation. To handle this situation, we generalize the single recourse action $\mathbf{u}_t^*$ in Problem Formulations 1 and 2 to a sequence of recourse actions

$$\mathbf{U}_t^* = (\mathbf{u}_t^*, \mathbf{u}_{t+1}^*, \cdots, \mathbf{u}_{t+M}^*).$$

Correspondingly, the recourse function is modified to adopt a sequence-to-sequence network architecture. The subsequent procedures, including the Abduction–Action–Prediction pipeline and backtracking counterfactual reasoning, remain conceptually the same and can be applied in this extended setting. A key challenge here is that the length of the anomaly sequence is not always known. As a practical solution, we propose computing the shortest sequence of recourse actions that adequately explains the abnormal states, i.e., the minimal sequence that restores normal states for all subsequent time steps. We describe the pseudo-code of the training process for recourse predictions with multiple abnormal time steps in Appendix A.

#### 4.5.2 Dealing with Anomalies due to Structural Intervention

We have assumed that anomalies are caused by external interventions on exogenous variables, without altering the Granger causal relationships. This type of anomaly occurs, for example, when a sensor is attacked. However, there are other types of anomalies that cannot be treated as external interventions but must be considered as structural interventions. In other words, they are caused by replacing the normal equations with abnormal ones. For instance, when a system experiences a structural change due to component degradation or failure. Despite the intrinsic differences between external interventions and structural interventions, it is still possible to explain anomalies caused by structural interventions using recourse actions. This is because we can rearrange the structural equation as follows: $x_t^{(j)} = \tilde{f}^{(j)}(\mathbf{x}_{\leq t-1}^{(1)}, \cdots, \mathbf{x}_{\leq t-1}^{(d)}) + u_t^{(j)} = f^{(j)}(\mathbf{x}_{\leq t-1}^{(1)}, \cdots, \mathbf{x}_{\leq t-1}^{(d)}) + u_t^{(j)} + \epsilon_t^{(j)}$, for $1 \leq j \leq d$, where $\tilde{f}^{(j)}(\mathbf{x}_{\leq t-1}^{(1)}, \cdots, \mathbf{x}_{\leq t-1}^{(d)})$ is an abnormal function for the time series $j$ at time $t$ and $\epsilon_t^{(j)} = \tilde{f}^{(j)}(\mathbf{x}_{\leq t-1}^{(1)}, \cdots, \mathbf{x}_{\leq t-1}^{(d)}) - f^{(j)}(\mathbf{x}_{\leq t-1}^{(1)}, \cdots, \mathbf{x}_{\leq t-1}^{(d)})$. As a result, we can describe the anomalies as

$$x_t^{(j)} := f^{(j)}(\mathbf{x}_{\leq t-1}^{(1)}, \cdots, \mathbf{x}_{\leq t-1}^{(d)}) + u_t^{(j)} + \epsilon_t^{(j)}, \text{ for } 1 \leq j \leq d, \tag{8}$$

where anomaly term $\epsilon_t^{(j)}$ can be due to a structural intervention. However, the significance of using recourse actions to explain structural intervention is worthy of future study.

## 5 Experiments

### 5.1 Experimental Setups

**Datasets**. We conduct experiments on two semi-synthetic datasets and one real-world dataset. The purposes of using semi-synthetic datasets are as follows. 1) We can derive the ground truth downstream time series after the intervention on the abnormal time step based on the data generation equations. 2) We can evaluate the fine-grained performance of RecAD by injecting different types of anomalies.

*Semi-synthetic datasets*: 1) **Linear Dataset** (Marcinkevičs & Vogt, 2021) is a time series dataset with linear interaction dynamics. 2) **Lotka-Volterra** is a nonlinear time series model that simulates a prairie ecosystem with multiple species. For both datasets, we adopt the structural equations defined in (Marcinkevičs & Vogt, 2021), which are included in Appendix B. We also describe the anomaly injection strategies in Appendix B.

*Real-world dataset*: **Multi-Source Distributed System (MSDS)** (Nedelkoski et al., 2020) consists of 10-dimensional time series. The first half of MSDS without anomalies is used as a training set, while the second half, including 5.37% abnormal time steps, is used as a test set. As a real-world dataset, we cannot observe the downstream time series after the intervention. In the test phase, we use GVAR and AAP to generate the counterfactual time series for evaluation.

Table 1: Statistics of three datasets for anomaly detection.

| Dataset | Dim. | Train | Test (Anomalies %) | | |
|---|---|---|---|---|---|
| | | | Point (External) | Seq. (External) | Seq. (Structural) |
| Linear | 4 | 50,000 | 250,000 (2%) | 250,000 (6%) | 250,000 (6%) |
| Lotka-Volterra | 20 | 100,000 | 500,000 (1%) | 500,000 (3%) | 500,000 (3%) |
| MSDS | 10 | 146,340 | 146,340 (5.37%) | | |

Table 1 shows the statistics of three datasets. Training datasets only consist of normal time series. Note that the test sets listed in Table 1 are used for evaluating the performance of anomaly detection. After detecting the abnormal time series in the test set, for the synthetic datasets, we use 50% of abnormal time series for training RecAD and another 50% for evaluating the performance of RecAD on recourse prediction, while for the MSDS dataset, we use 80% of the abnormal time series for training RecAD and the rest 20% for evaluation.

**Baselines**. To our best knowledge, there is no causal algorithmic recourse approach in time series anomaly detection. We compare RecAD with the following baselines: 1) Multilayer perceptron (MLP), which is trained with the normal flattened sliding windows to predict the normal values for the next step; 2) LSTM, which can learn complex temporal dependencies in time series to make predictions for the next step; 3) Vector Autoregression (VAR) is a statistical model that used to analyze GC within multivariate time series data and predict future values; 4) Generalised Vector Autoregression (GVAR) (Marcinkevičs & Vogt, 2021) is an extension of self-explaining neural network that can infer nonlinear multivariate GC and predict values of the next step.

For all the baselines, in the training phase, we train them to predict the last value in a time window on the normal time series. In the testing phase, when a time window is detected as abnormal by an anomaly detection model, indicating the last time step $\mathbf{x}_t$ is abnormal, we use baselines to predict the expected normal value in the last time step $\tilde{\mathbf{x}}_t$. Then, the recourse action values can be derived as $\boldsymbol{\theta}_t = \tilde{\mathbf{x}}_t - \mathbf{x}_t$. For the sequence anomalies, we keep using the baselines to predict the expected normal values and derive the action values by comparing them with the observed values.

**Evaluation Metrics**. We evaluate the performance of algorithmic recourse in terms of effectiveness, feasibility, and efficiency.

1) **Flipping Ratio ($\uparrow$) (Effectiveness)**. The Flipping Ratio measures the proportion of detected anomalous time steps that are successfully reverted to normal after applying the recourse actions. Formally, let $\mathcal{T}_{\mathrm{anom}}$ be the set of detected anomalous time steps, let $\boldsymbol{\theta}_t$ be the recourse actions, $\tilde{\mathbf{x}}$ the counterfactual sequence produced by the recourse algorithm, and $s(\cdot)$ the anomaly score with threshold $\tau$. Then

$$\text{Flipping Ratio} = \frac{\sum_{t \in \mathcal{T}_{\mathrm{anom}}} \mathbb{I}\{s(\tilde{\mathbf{x}}_t) < \tau\}}{|\mathcal{T}_{\mathrm{anom}}|}.$$

A higher Flipping Ratio indicates more effective recourse, as a larger fraction of anomalies is eliminated after intervention.

2) **Action Cost ($\downarrow$) (Feasibility)**. The Action Cost quantifies the magnitude of all interventions required to flip anomalies. For each time step $t$, let $\theta_t \in \mathbb{R}^d$ denote the perturbation applied to the $d$-dimensional system state. The cost at time $t$ is measured using the Euclidean norm $\|\theta_t\|_2$, and the average cost per abnormal

sequence is computed as

$$\text{Action Cost} = \frac{1}{|\mathcal{T}_{\text{anom}}|} \sum_{t \in \mathcal{T}_{\text{anom}}} \|\theta_t\|_2^2.$$

Lower Action Cost reflects more parsimonious interventions that achieve anomaly flipping with minimal total perturbation.

3) **Action Step (↓) (Efficiency).** The Action Step metric counts the number of discrete time steps at which interventions are applied. For each anomalous sequence, let $s_t = 1$ if $\theta_t \neq 0$ (i.e., an action is taken) and $s_t = 0$ otherwise. The average number of steps per sequence is

$$\text{Action Step} = \frac{1}{|\mathcal{T}_{\text{anom}}|} \sum_{t \in \mathcal{T}_{\text{anom}}} s_t.$$

A smaller value implies that fewer intervention points are required, indicating temporally efficient recourse.

## 5.2 Experimental Results

To implement the score function $s(\cdot)$, in our experiments, we leverage the UnSupervised Anomaly Detection for multivariate time series (USAD) (Audibert et al., 2020) as well as a transformer-based anomaly detection model (TranAD) (Tuli et al., 2022) as base anomaly detection models that output an anomaly score for multivariate time series. We include the implementation details as well as the performance of USAD and TranAD for anomaly detection in the Appendix. For all experimental results, we report the mean and standard deviation over 10 runs. Our code is available online[1].

### 5.2.1 Evaluation Results on Synthetic Datasets

**The performance of recourse prediction on anomalies caused by external interventions.** Table 2 shows the performance of RecAD for recourse prediction on anomalies caused by external interventions. Note that the anomalies consist of both point and sequential anomalies. First, in all settings, RecAD can achieve the highest flipping ratios, which shows the effectiveness of RecAD in flipping abnormal behavior. Meanwhile, RecAD can achieve low action costs and action steps compared with other baselines. Because no baseline considers the downstream impact of recourse actions, they usually require more action steps to flip the anomalies.

**The performance of recourse prediction on anomalies caused by structural interventions.** We examine the performance of RecAD on anomalies caused by structural intervention. As shown in Table 3, RecAD can achieve the highest flipping ratio compared with baselines on both Linear and Lotka-Volterra datasets, indicating that the majority of anomalies caused by structural interventions can be successfully flipped. Meanwhile, RecAD can also achieve low action costs and action steps with high flipping ratios. Overall, RecAD meets the requirement of algorithmic recourse, i.e., flipping the abnormal outcome with minimum costs on anomalies caused by structural interventions.

Therefore, based on Tables 2 and 3, we can demonstrate that RecAD can provide recourse prediction on different types of anomalies in multivariate time series.

### 5.2.2 Evaluation Results on the MSDS Dataset

**The performance of recourse prediction.** Because the types of anomalies in the real-world dataset are unknown, we report the performance of recourse prediction on any detected anomalies. As shown in Table 4, RecAD achieves much higher flipping ratios than all baselines. Regarding the average action cost per time series and the average action step, RecAD also outperforms the baselines by having the lowest values. This suggests that, by incorporating Granger causality, RecAD can identify recourse actions that minimize both cost and the number of steps.

---

[1] https://github.com/hanxiao0607/RecAD

Table 2: The performance of recourse prediction on anomalies caused by external interventions.

(a) USAD as the anomaly detection model

| Dataset | Model | Point | | | Seq. | | |
|---|---|---|---|---|---|---|---|
| | | Flipping Ratio ↑ | Action Cost ↓ | Action Step ↓ | Flipping Ratio ↑ | Action Cost ↓ | Action Step ↓ |
| Linear | MLP | $0.778_{\pm0.054}$ | $8.406_{\pm0.257}$ | $1.188_{\pm0.051}$ | $0.867_{\pm0.048}$ | $22.394_{\pm0.545}$ | $2.261_{\pm0.027}$ |
| | LSTM | $0.807_{\pm0.045}$ | $8.383_{\pm0.253}$ | $1.170_{\pm0.040}$ | $0.878_{\pm0.044}$ | $22.439_{\pm0.553}$ | $2.248_{\pm0.031}$ |
| | VAR | $0.676_{\pm0.063}$ | $8.841_{\pm0.224}$ | $1.311_{\pm0.077}$ | $0.765_{\pm0.061}$ | $23.696_{\pm0.511}$ | $2.434_{\pm0.037}$ |
| | GVAR | $0.775_{\pm0.053}$ | $8.446_{\pm0.249}$ | $1.207_{\pm0.052}$ | $0.848_{\pm0.051}$ | $22.415_{\pm0.547}$ | $2.287_{\pm0.029}$ |
| | RecAD | $\mathbf{0.901}_{\pm\mathbf{0.035}}$ | $\mathbf{8.201}_{\pm\mathbf{0.176}}$ | $\mathbf{1.104}_{\pm\mathbf{0.024}}$ | $\mathbf{0.944}_{\pm\mathbf{0.041}}$ | $\mathbf{21.264}_{\pm\mathbf{0.947}}$ | $\mathbf{2.193}_{\pm\mathbf{0.046}}$ |
| Lotka-Volterra | MLP | $0.741_{\pm0.126}$ | $277.580_{\pm117.126}$ | $1.237_{\pm0.180}$ | $0.688_{\pm0.259}$ | $761.550\pm64.422$ | $2.195_{\pm0.757}$ |
| | LSTM | $0.893_{\pm0.087}$ | $313.510_{\pm124.921}$ | $1.096_{\pm0.061}$ | $0.889_{\pm0.097}$ | $1590.483_{\pm85.126}$ | $1.339_{\pm0.145}$ |
| | VAR | $0.558_{\pm0.166}$ | $326.967_{\pm126.322}$ | $1.57_{\pm0.324}$ | $0.504_{\pm0.151}$ | $1445.084_{\pm189.943}$ | $2.570_{\pm0.676}$ |
| | GVAR | $0.493_{\pm0.332}$ | $270.335_{\pm117.223}$ | $2.020_{\pm1.049}$ | $0.606_{\pm0.266}$ | $\mathbf{749.792}_{\pm\mathbf{105.656}}$ | $2.547_{\pm0.905}$ |
| | RecAD | $\mathbf{0.915}_{\pm\mathbf{0.088}}$ | $\mathbf{262.759}_{\pm\mathbf{99.008}}$ | $\mathbf{1.085}_{\pm\mathbf{0.055}}$ | $\mathbf{0.972}_{\pm\mathbf{0.016}}$ | $1374.112_{\pm343.470}$ | $\mathbf{1.329}_{\pm\mathbf{0.192}}$ |

(b) TranAD as the anomaly detection model

| Dataset | Model | Point | | | Seq. | | |
|---|---|---|---|---|---|---|---|
| | | Flipping Ratio ↑ | Action Cost ↓ | Action Step ↓ | Flipping Ratio ↑ | Action Cost ↓ | Action Step ↓ |
| Linear | MLP | $0.657_{\pm0.217}$ | $\mathbf{5.260}_{\pm\mathbf{2.177}}$ | $1.145_{\pm0.413}$ | $0.732_{\pm0.228}$ | $\mathbf{14.006}_{\pm\mathbf{6.043}}$ | $2.238_{\pm0.228}$ |
| | LSTM | $0.671_{\pm0.246}$ | $5.271_{\pm2.123}$ | $1.441_{\pm0.472}$ | $0.759_{\pm0.234}$ | $14.228_{\pm5.953}$ | $2.189_{\pm0.184}$ |
| | VAR | $0.557_{\pm0.210}$ | $5.349_{\pm2.278}$ | $1.551_{\pm0.365}$ | $0.642_{\pm0.227}$ | $14.147_{\pm6.202}$ | $2.373_{\pm0.228}$ |
| | GVAR | $0.590_{\pm0.227}$ | $5.326_{\pm2.241}$ | $1.549_{\pm0.537}$ | $0.673_{\pm0.237}$ | $14.211_{\pm6.038}$ | $2.335_{\pm0.308}$ |
| | RecAD | $\mathbf{0.884}_{\pm\mathbf{0.121}}$ | $7.021_{\pm1.471}$ | $\mathbf{1.112}_{\pm\mathbf{0.046}}$ | $\mathbf{0.924}_{\pm\mathbf{0.115}}$ | $16.793_{\pm5.600}$ | $\mathbf{1.905}_{\pm\mathbf{0.400}}$ |
| Lotka-Volterra | MLP | $0.796_{\pm0.194}$ | $253.537_{\pm115.642}$ | $1.230_{\pm0.246}$ | $0.735_{\pm0.153}$ | $826.481_{\pm32.198}$ | $2.251_{\pm0.448}$ |
| | LSTM | $0.839_{\pm0.190}$ | $274.916_{\pm119.687}$ | $1.146_{\pm0.173}$ | $0.833_{\pm0.099}$ | $1457.314_{\pm34.293}$ | $1.866_{\pm0.214}$ |
| | VAR | $0.636_{\pm0.293}$ | $286.372_{\pm116.162}$ | $1.554_{\pm0.583}$ | $0.420_{\pm0.098}$ | $1261.513_{\pm103.780}$ | $2.833_{\pm0.399}$ |
| | GVAR | $0.720_{\pm0.349}$ | $\mathbf{244.942}_{\pm\mathbf{114.734}}$ | $1.664_{\pm1.052}$ | $0.550_{\pm0.149}$ | $\mathbf{680.181}_{\pm\mathbf{44.329}}$ | $2.501_{\pm0.611}$ |
| | RecAD | $\mathbf{0.902}_{\pm\mathbf{0.074}}$ | $278.432_{\pm127.007}$ | $\mathbf{1.099}_{\pm\mathbf{0.070}}$ | $\mathbf{0.925}_{\pm\mathbf{0.034}}$ | $1254.646_{\pm196.460}$ | $\mathbf{1.856}_{\pm\mathbf{0.138}}$ |

Table 3: The performance of recourse prediction on anomalies caused by structural interventions.

(a) USAD as the anomaly detection model

| Dataset | Model | Flipping Ratio ↑ | Action Cost ↓ | Action Step ↓ |
|---|---|---|---|---|
| Linear | MLP | $0.884_{\pm0.023}$ | $41.387_{\pm2.151}$ | $2.359_{\pm0.035}$ |
| | LSTM | $0.903_{\pm0.021}$ | $42.412_{\pm2.002}$ | $2.398_{\pm0.043}$ |
| | VAR | $0.782_{\pm0.035}$ | $59.346_{\pm3.285}$ | $2.883_{\pm0.062}$ |
| | GVAR | $0.874_{\pm0.024}$ | $39.474_{\pm2.116}$ | $2.415_{\pm0.041}$ |
| | RecAD | $\mathbf{0.919}_{\pm\mathbf{0.037}}$ | $\mathbf{38.917}_{\pm\mathbf{6.247}}$ | $\mathbf{2.169}_{\pm\mathbf{0.206}}$ |
| Lotka-Volterra | MLP | $0.665_{\pm0.277}$ | $\mathbf{1578.597}_{\pm\mathbf{49.253}}$ | $2.247_{\pm0.744}$ |
| | LSTM | $0.890_{\pm0.099}$ | $3159.333_{\pm173.463}$ | $2.310_{\pm0.100}$ |
| | VAR | $0.488_{\pm0.178}$ | $3088.788_{\pm435.034}$ | $2.630_{\pm0.640}$ |
| | GVAR | $0.584_{\pm0.277}$ | $1846.415_{\pm347.144}$ | $2.618_{\pm0.858}$ |
| | RecAD | $\mathbf{0.970}_{\pm\mathbf{0.012}}$ | $2767.819_{\pm581.046}$ | $\mathbf{1.386}_{\pm\mathbf{0.120}}$ |

(b) TranAD as the anomaly detection model

| Dataset | Model | Flipping Ratio ↑ | Action Cost ↓ | Action Step ↓ |
|---|---|---|---|---|
| Linear | MLP | $0.834_{\pm0.117}$ | $52.812_{\pm37.404}$ | $2.722_{\pm0.652}$ |
| | LSTM | $0.921_{\pm0.031}$ | $52.759_{\pm31.272}$ | $2.638_{\pm0.417}$ |
| | VAR | $0.778_{\pm0.083}$ | $61.716_{\pm37.610}$ | $2.908_{\pm0.421}$ |
| | GVAR | $0.785_{\pm0.090}$ | $59.303_{\pm38.320}$ | $2.840_{\pm0.430}$ |
| | RecAD | $\mathbf{0.942}_{\pm\mathbf{0.046}}$ | $\mathbf{48.227}_{\pm\mathbf{31.905}}$ | $\mathbf{2.260}_{\pm\mathbf{0.447}}$ |
| Lotka-Volterra | MLP | $0.727_{\pm0.167}$ | $1541.437_{\pm65.036}$ | $2.228_{\pm0.504}$ |
| | LSTM | $0.853_{\pm0.091}$ | $2783.639_{\pm121.668}$ | $\mathbf{1.735}_{\pm\mathbf{0.176}}$ |
| | VAR | $0.444_{\pm0.105}$ | $2441.456_{\pm244.908}$ | $2.791_{\pm0.428}$ |
| | GVAR | $0.572_{\pm0.171}$ | $\mathbf{1387.077}_{\pm\mathbf{118.036}}$ | $2.495_{\pm0.614}$ |
| | RecAD | $\mathbf{0.929}_{\pm\mathbf{0.035}}$ | $2276.269_{\pm481.183}$ | $1.809_{\pm0.145}$ |

Table 4: The performance of recourse prediction in MSDS datasets.

(a) USAD as anomaly detection model

| Model | Flipping Ratio ↑ | Action Cost ↓ | Action Step ↓ |
|---|---|---|---|
| MLP | $0.687_{\pm 0.282}$ | $6.848_{\pm 2.506}$ | $1.443_{\pm 0.680}$ |
| LSTM | $0.830_{\pm 0.211}$ | $6.798_{\pm 2.604}$ | $1.279_{\pm 0.493}$ |
| VAR | $0.704_{\pm 0.273}$ | $6.759_{\pm 2.821}$ | $1.432_{\pm 0.596}$ |
| GVAR | $0.712_{\pm 0.211}$ | $8.923_{\pm 3.258}$ | $1.425_{\pm 0.466}$ |
| RecAD | $\mathbf{0.841_{\pm 0.080}}$ | $\mathbf{6.747_{\pm 1.543}}$ | $\mathbf{1.249_{\pm 0.068}}$ |

(b) TranAD as anomaly detection model

| Model | Flipping Ratio ↑ | Action Cost ↓ | Action Step ↓ |
|---|---|---|---|
| MLP | $0.664_{\pm 0.373}$ | $4.999_{\pm 3.136}$ | $1.470_{\pm 0.891}$ |
| LSTM | $0.749_{\pm 0.257}$ | $5.443_{\pm 3.011}$ | $1.301_{\pm 0.606}$ |
| VAR | $0.643_{\pm 0.366}$ | $5.197_{\pm 3.118}$ | $1.525_{\pm 0.825}$ |
| GVAR | $0.739_{\pm 0.232}$ | $6.443_{\pm 3.270}$ | $1.327_{\pm 0.488}$ |
| RecAD | $\mathbf{0.837_{\pm 0.159}}$ | $\mathbf{4.918_{\pm 1.292}}$ | $\mathbf{1.273_{\pm 0.453}}$ |

## 5.3 Case Study

We further conduct case studies to show how to use the recourse action predicted by RecAD as an explanation for anomaly detection in multivariate time series.

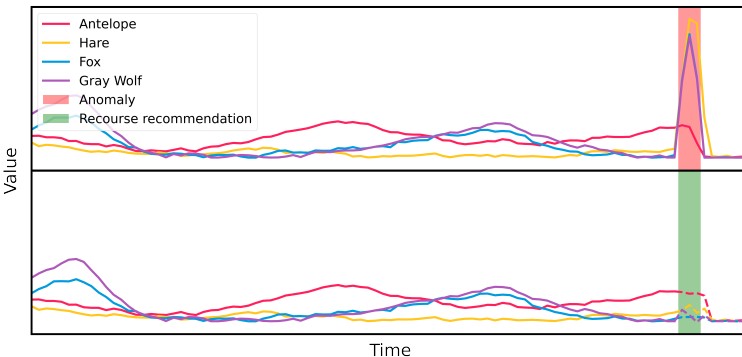

Figure 3: Recourse recommendations for intervening in an imbalanced ecosystem to restore balance.

**Case study on the Lotka-Volterra dataset.** Figure 3 shows a simulation of a prairie ecosystem that contains antelope, hare, fox, and gray wolf based on the Lotka-Volterra model (Bacaër, 2011), where each time series indicates the population of a species. As shown in the top figure, in most of the time steps, the numbers of carnivores (fox and gray wolf) and herbivores (antelope and hare) keep stable in a balanced ecosystem, say 0.1k-1k antelopes, 1k-10k hares, 0.1k-1k foxes, and 0.1k-1k gray wolves. After detecting abnormal behavior at a specific time step (red area in the top figure), the algorithmic recourse aims to provide recourse actions to flip the abnormal outcome. In this case, the algorithmic recourse model recommends the intervention of reducing the populations of hares, foxes, and gray wolves by 100.1k, 9.3k, and 7.5k, respectively. After applying the recourse actions (green area in the bottom figure), we can notice the populations of four species become stable again (the dashed line in the bottom figure). Therefore, the recourse actions can provide recommendations to restore the balance of the prairie ecosystem.

**Case study on the MSDS dataset.** Figure 4 depicts a case study on MSDS with control nodes 117 and 124. USAD detects a subsequence of anomaly consisting of two abnormal time steps from two-time series (CPU and RAM usages on node 117), highlighted in the red area of the top figure.

When the first abnormal time step is detected, RecAD suggests releasing the CPU usage by 6.7 on node 117 (the green area in the middle figure). In other words, it also means the anomaly here is due to the higher CPU usage than normal with a value of 6.7. After taking this action, the following time steps are affected by this action. A counterfactual time series is then generated using the AAP process, which is shown as the dashed lines in the middle of Figure 4. RecAD continues to monitor subsequent time steps for any abnormalities.

The following time step is still detected as abnormal in the time series of memory usage of node 117. RecAD recommends releasing the RAM usage by 13.39 on node 117 (the green area in the bottom figure), meaning that the abnormal time step here is due to high memory usage in a margin of 13.39. After taking the recourse action, the counterfactual time series is then generated (the dashed lines in the bottom figure). We can then observe that the entire time series returns to normal.

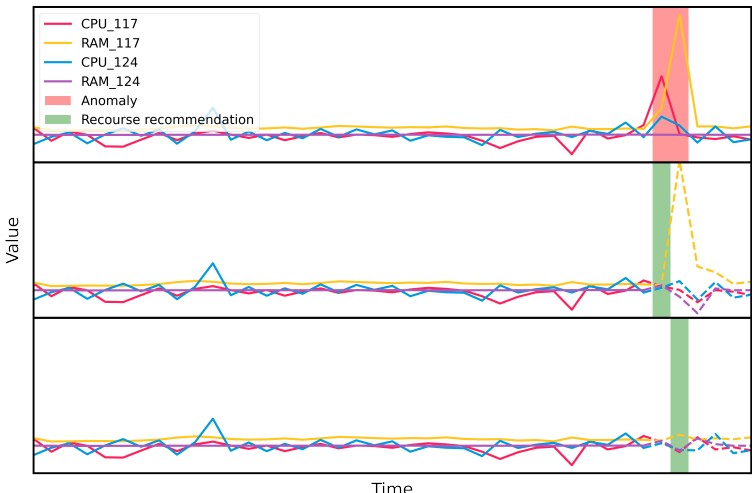

Figure 4: Recourse recommendations for restoring the abnormal CPU and RAM usages in MSDS.

In summary, recourse actions recommended by RecAD can effectively flip the outcome and lead to a normal counterfactual time series. Meanwhile, based on the recourse actions, the domain expert can understand why a time step is abnormal.

Table 5: The performance of recourse prediction using different components of RecAD.

| Dataset | Metric | RecAD w/o FFNN | RecAD w/o LSTM | RecAD |
|---|---|---|---|---|
| Linear | Flipping Ratio ↑ | $0.340_{\pm 0.191}$ | $0.676_{\pm 0.085}$ | $\mathbf{0.922_{\pm 0.040}}$ |
| | Action Cost ↓ | $119.286_{\pm 174.173}$ | $23.589_{\pm 23.453}$ | $\mathbf{22.794_{\pm 13.054}}$ |
| | Action Step ↓ | $2.905_{\pm 0.882}$ | $2.276_{\pm 0.939}$ | $\mathbf{1.822_{\pm 0.521}}$ |
| Lotka-Volterra | Flipping Ratio ↑ | $0.353_{\pm 0.210}$ | $0.523_{\pm 0.090}$ | $\mathbf{0.952_{\pm 0.054}}$ |
| | Action Cost ↓ | $\mathbf{876.107_{\pm 810.047}}$ | $1035.192_{\pm 881.242}$ | $1468.230_{\pm 1086.090}$ |
| | Action Step ↓ | $2.706_{\pm 1.068}$ | $2.304_{\pm 0.646}$ | $\mathbf{1.266_{\pm 0.180}}$ |
| MSDS | Flipping Ratio ↑ | $0.228_{\pm 0.147}$ | $0.697_{\pm 0.214}$ | $\mathbf{0.841_{\pm 0.080}}$ |
| | Action Cost ↓ | $\mathbf{3.494_{\pm 2.665}}$ | $4.136_{\pm 0.727}$ | $6.747_{\pm 1.543}$ |
| | Action Step ↓ | $2.048_{\pm 0.607}$ | $1.581_{\pm 0.525}$ | $\mathbf{1.249_{\pm 0.068}}$ |

## 5.4 Ablation Study

We evaluate the performance of using different parts of RecAD (i.e., FFNN and LSTM) for recourse prediction. In this experiment, we adopt USAD as the base anomaly detection model. As RecAD contains an LSTM to catch the previous $K-1$ time lags and a feedforward neural network (FFNN) to include the time lag exclusion term $\Delta_t$, we then test the performance of these two parts separately. Table 5 shows the average flipping ratio, action cost, and action step for three types of anomalies for the synthetic datasets and results for the real-world dataset MSDS. We observe that RecAD achieves higher flipping ratios while requiring fewer or comparable action steps compared to methods that utilize only parts of RecAD. In cases where other methods exhibit lower action costs, they achieve very low flipping ratios, indicating their inability to successfully flip abnormal samples, which is undesired. This result shows the importance of considering both information for reasonable action value prediction.

## 5.5 Sensitivity Analysis

The objective function defined in Eq. (7) for training RecAD employs the hyperparameter $\lambda$ to balance the flipping ratio and action cost. A large $\lambda$ indicates a large penalty for high action costs, which could potentially hurt the performance of flipping abnormal time steps as small action costs may not be sufficient to flip the anomalies. As shown in Figure 5, on both synthetic datasets, we have similar observations that with the increase of $\lambda$, both action cost and flipping ratio decrease. Specifically, smaller values of $\lambda$ yield higher

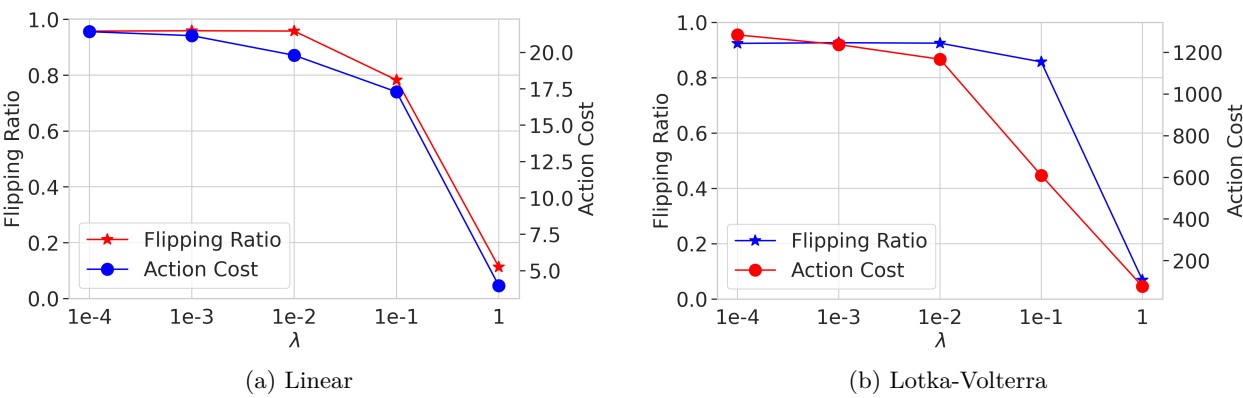

(a) Linear

(b) Lotka-Volterra

Figure 5: Effects of the hyperparameter $\lambda$ in Eq. (7).

action costs but higher flipping ratios, whereas larger values produce lower costs and lower flipping ratios. By varying $\lambda$, the model generates multiple distinct recourse trajectories with different cost-effectiveness trade-offs, introducing diversity among the counterfactuals.

## 6 Conclusions

In this work, we have developed RecAD, a novel framework for algorithmic recourse in abnormal multivariate time series. RecAD suggests actionable interventions to restore the multivariate time series to normal status, offering counterfactual explanations for abnormal patterns. By leveraging backtracking counterfactual reasoning, we formulated the problem of learning the recourse function from data as a constrained maximum likelihood problem to enable end-to-end training. The empirical studies have demonstrated the effectiveness of RecAD for recommending explainable and practical recourse actions in abnormal time series.

## 7 Limitations and Future Work

Our current framework adopts GVAR to infer Granger causal relationships from multivariate time series. While GVAR provides a principled and computationally tractable approach, several limitations remain: (i) GVAR may be sensitive to latent confounding and hyperparameter choices governing sparsity and temporal smoothness. (ii) While GVAR can model nonlinearity and time variation, its reliability still depends on regularization design and stability selection thresholds when aggregating coefficients into a summary graph. (iii) GVAR does not explicitly handle interventions or invariance across environments, which can improve robustness to spurious associations. In future work, we plan to integrate recent advances in causal discovery to complement GVAR's learned coefficients, including invariance-based or multi-environment selection to reduce spurious edges, procedures that address latent confounders and non-stationarity, and hybrid models that combine constraint-based, score-based, and neural approaches.

Another limitation of the current work lies in handling anomalies caused by structural interventions. In Section 4.5.2, we reformulated structural interventions as external interventions and addressed them using a similar method. However, this approach does not directly capture or explain the structural changes themselves. As future work, we plan to develop methods that can learn a modified causal structure, which, if implemented, would restore the system to normal behavior. In particular, we aim to design an optimization procedure that enables the encoder-decoder architecture to adapt rapidly to local causal mechanism changes present in abnormal sequences.

## Acknowledgment

This work was supported in part by NSF 1910284, 2103829, and 2142725.

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

## Appendix

## A    Algorithm of RecAD

Algorithm 1 shows the pseudo-code of the training process for recourse predictions. Given an abnormal time window $\mathbf{W}_t$ with the abnormal time step $t$ and a subsequence window $\mathbf{V}_{t+L}$ of length $L$, we first predict the action value on $\mathbf{x}_t$ to reverse the abnormal status. After conducting the recourse actions, we also need to ensure that the following $L$ time steps are also normal. Therefore, we then derive the counterfactual subsequence $\mathbf{V}_{t+L}^*$ step by step. For each following step, denoted as $\mathbf{V}_{t+l}^*$, if the subsequence $\mathbf{V}_{t+l}^*$ is still abnormal, we further conduct recourse actions to flip the abnormal status. After going through $L$ time steps, we update the action prediction function $h_\phi(\cdot)$ based on the objective function defined in Eq. (7).

---

**Algorithm 1:** Pseudo code of training RecAD

    **Input:** Pretrained GVAR $g_k(\cdot)$, anomaly detector $s(\cdot)$, an abnormal window $\mathbf{W}_t$ with the abnormal time step $t$, and the following window $\mathbf{V}_{t+L}$
    **Output:** Updated $h_\phi(\cdot)$
  1: $\mathbf{x}_t^* \leftarrow$ Action_Prediction($\mathbf{W}_t$, $g_k(\cdot)$, $h_\phi(\cdot)$)
  2: $\mathbf{V}_{t+L}^*[0] = \mathbf{x}_t^*$
  3: **for** $l \leftarrow 1$ **to** $L$ **do**
  4:     Derive $\mathbf{V}_{t+l}^*$ by Eqs. (5) and (6)
  5:     **if** $s(\mathbf{V}_{t+l}^*) > \tau$ **then**
  6:         $\mathbf{x}_{t+l}^* \leftarrow$ Action_Prediction($\mathbf{V}_{t+l}^*$, $g_k(\cdot)$, $h_\phi(\cdot)$)
  7:         $\mathbf{V}_{t+L}^*[l] = \mathbf{x}_{t+l}^*$
  8:     **end if**
  9: **end for**
10: Update $h_\phi(\cdot)$ based on the objective function $\mathcal{L}(\phi)$ in Eq. (7)
11: **Function** Action_Prediction($\mathbf{W}_t$, $g_k(\cdot)$, $h_\phi(\cdot)$)
12:     $\mathbf{W}_{t-1} \leftarrow \mathbf{W}_t \setminus \{\mathbf{x}_t\}$
13:     Compute $\hat{\mathbf{x}}_t = \sum_{k=1}^{K-1} g_k(\mathbf{x}_{t-k})\mathbf{x}_{t-k}$ with $\mathbf{W}_{t-1}$
14:     Compute $\Delta_t = \mathbf{x}_t - \hat{\mathbf{x}}_t$
15:     Compute $\boldsymbol{\theta}_t = h_\phi(\mathbf{W}_{t-1}, \Delta_t)$
16:     $\mathbf{x}_t^* = \mathbf{x}_t + \boldsymbol{\theta}_t$
17:     **Return** $\mathbf{x}_t^*$

---

## B    Experiments

### B.1    Datasets

**Linear Dataset** (Marcinkevičs & Vogt, 2021) is a **synthetic** time series dataset with linear interaction dynamics. We follow the structure from the original paper (Marcinkevičs & Vogt, 2021), where the linear interaction contains no instantaneous effects between time series and can be defined as:

$$
\begin{aligned}
x_t^{(1)} &= a_1 x_{t-1}^{(1)} + u_t^{(1)} + \epsilon_t^{(1)}, \\
x_t^{(2)} &= a_2 x_{t-1}^{(2)} + a_3 x_{t-1}^{(1)} + u_t^{(2)} + \epsilon_t^{(2)}, \\
x_t^{(3)} &= a_4 x_{t-1}^{(3)} + a_5 x_{t-1}^{(2)} + u_t^{(3)} + \epsilon_t^{(3)}, \\
x_t^{(4)} &= a_6 x_{t-1}^{(4)} + a_7 x_{t-1}^{(2)} + a_8 x_{t-1}^{(3)} + u_t^{(4)} + \epsilon_t^{(4)},
\end{aligned}
\tag{9}
$$

where coefficients $a_i \sim \mathcal{U}([-0.8, -0.2] \cup [0.2, 0.8])$, additive innovation terms $u_t^{(\cdot)} \sim \mathcal{N}(0, 0.16)$, and anomaly term $\epsilon_t^{(\cdot)}$.

*Abnormal behavior injection.* For point anomalies, the anomaly terms are single or multiple extreme values for randomly selected time series variables at a specific time step $t$. For example, a point anomaly at time

step $t$ can be generated with an abnormal term $\boldsymbol{\epsilon}_t = [0, 2, 4, 0]$, which means the second and third time series have extreme values.

For non-causal sequence anomalies, the anomaly terms are function-generated values for a given time range from $t$ to $t + n$. For instance, setting $\epsilon_{t+i}^{(1)} = 0.1 \times i$, for $0 \le i \le n$, will cause a trend anomaly for time series variable $x$; setting $\epsilon_{t+i}^{(1)} \sim \mathcal{N}(0, 0.16)$, for $0 \le i \le n$, will cause a shapelet anomaly for time series variable $x^{(1)}$; and setting $\epsilon_{t+i}^{(1)} = (a_1 x_{t+2i-1}^{(1)} + u_{t+2i}^{(1)}) + (a_1 x_{t+2i-2}^{(1)} + u_{t+2i-1}^{(1)}) - (a_1 x_{t+i-1}^{(1)} + u_{t+i}^{(1)})$, for $0 \le i \le n$, will cause a seasonal anomaly for time series variable $x^{(1)}$.

For causal sequence anomalies, we consider two scenarios: 1) changing the coefficients $\mathcal{A} = \{a_1, a_2, \cdots, a_8\}$ from a normal one to a different one in a time range $t$ to $t + n$; 2) changing generative functions from the original Equation (9) to a different one.

**Lotka-Volterra** (Bacaër, 2011) is another **synthetic** time series model that simulates a prairie ecosystem with multiple species. We follow the structure from (Marcinkevičs & Vogt, 2021), which defines as:

$$
\begin{aligned}
\frac{d\mathbf{x}^{(i)}}{dt} &= \alpha \mathbf{x}^{(i)} - \beta \sum_{j \in Pa(\mathbf{x}^{(i)})} \mathbf{y}^{(j)} - \eta (\mathbf{x}^{(i)})^2, \text{ for } 1 \le j \le p, \\
\frac{d\mathbf{y}^{(j)}}{dt} &= \delta \mathbf{y}^{(j)} \sum_{k \in Pa(\mathbf{y}^{(j)})} \mathbf{x}^{(k)} - \rho \mathbf{y}^{(j)}, \text{ for } 1 \le j \le p, \\
x_t^{(i)} &= x_t^{(i)} + \epsilon_t^{(i)}, \text{ for } 1 \le j \le p, \\
y_t^{(j)} &= y_t^{(j)} + \epsilon_t^{(j)}, \text{ for } 1 \le j \le p,
\end{aligned}
\tag{10}
$$

where $\mathbf{x}^{(i)}$ and $\mathbf{y}^{(j)}$ denote the population sizes of prey and predator, respectively; $\alpha, \beta, \eta, \delta, \rho$ are parameters that decide the strengths of interactions, $Pa(\mathbf{x}^{(i)})$ and $Pa(\mathbf{y}^{(j)})$ correspond the Granger Causality between prey and predators for $\mathbf{x}^{(i)}$ and $\mathbf{y}^{(j)}$ respectively, and $\epsilon_t^{(\cdot)}$ is the abnormal term. We adopt 10 prey species and 10 predator species.

*Abnormal behavior injection.* For point anomalies and non-causal sequence anomalies, we perform a similar procedure as the linear dataset, i.e., randomly select time series variables at a specific time step $t$ and assign single or multiple extreme values as point anomalies, and assign function-generated abnormal terms for a time range from $t$ to $t + n$ as sequence anomalies.

For causal sequence anomalies, we still consider two scenarios: 1) changing the coefficients $\alpha, \beta, \eta, \delta, \rho$ to different values than the normal ones; 2) changing $Pa(\mathbf{x}^{(i)})$ and $Pa(\mathbf{y}^{(j)})$ to different ones from the original generative functions Equation (10).

## B.2 Implementation Details

Similar to (Audibert et al., 2020), we adopt a sliding window with sizes 5, 5, and 10 for the Linear, Lotka-Volterra, and MSDS datasets, respectively. We set the hyperparameters for GVAR by following (Marcinkevičs & Vogt, 2021). When training $h_\phi(\cdot)$, we set $L$ in the objective function as $L = 1$, which is to ensure the following one-time step should be normal. The cost vector $\mathbf{c}$ can be changed according to the requirements or prior knowledge. Because the baseline models are prediction-based models that cannot take the cost into account, to be fair, we use $\mathbf{1}$ as the cost vector. The implementation details of neural networks in experiments are described in the Appendix.

For baselines, MLP is a feed-forward neural network with a structure of $((K - 1) * d)$-100-100-100-$d$ that the input is the flattened vector of $K - 1$ time steps with $d$ dimensions and the output is the predicted value of the next time step. The LSTM model consists of one hidden layer with 100 dimensions and is connected with a fully connected layer with a structure of 100-$d$. We use statsmodels[2] to implement the VAR model. The baseline GVAR model is the same as GVAR within our framework. To implement $h_\phi(\cdot)$ in RecAD, we utilize

---

[2]https://www.statsmodels.org/

an LSTM that consists of one hidden layer with 100 dimensions and a feed-forward network with structure $d$-100. Then we use another feed-forward network with a structure of 200-$d$ to predict the intervention values.

All experiments were conducted on an Ubuntu 20.04 server equipped with an AMD Ryzen 3960X 24-Core processor at 3.8GHz, dual GeForce RTX 3090 GPUs, and 128 GB of RAM. The implementation uses Python 3.9.7 and PyTorch 1.11.0.

### B.3 Performance of anomaly detection.

Table 6: Performance of Anomaly Detection: TranAD and USAD on Synthetic Datasets

| Anomaly Types | Metrics | Linear | | Lotka-Volterra | |
|---|---|---|---|---|---|
| | | TranAD | USAD | TranAD | USAD |
| Non-causal Point | F1 | $0.736_{\pm 0.058}$ | $0.749_{\pm 0.022}$ | $0.857_{\pm 0.020}$ | $0.787_{\pm 0.106}$ |
| | AUC-PR | $0.602_{\pm 0.065}$ | $0.619_{\pm 0.024}$ | $0.742_{\pm 0.033}$ | $0.840_{\pm 0.116}$ |
| | AUC-ROC | $0.812_{\pm 0.038}$ | $0.816_{\pm 0.016}$ | $0.950_{\pm 0.002}$ | $0.851_{\pm 0.068}$ |
| Non-causal Seq. | F1 | $0.829_{\pm 0.093}$ | $0.878_{\pm 0.011}$ | $0.783_{\pm 0.032}$ | $0.677_{\pm 0.061}$ |
| | AUC-PR | $0.730_{\pm 0.123}$ | $0.798_{\pm 0.015}$ | $0.633_{\pm 0.054}$ | $0.519_{\pm 0.026}$ |
| | AUC-ROC | $0.891_{\pm 0.066}$ | $0.914_{\pm 0.010}$ | $0.879_{\pm 0.013}$ | $0.794_{\pm 0.089}$ |
| Causal Seq. | F1 | $0.740_{\pm 0.043}$ | $0.756_{\pm 0.003}$ | $0.747_{\pm 0.025}$ | $0.714_{\pm 0.020}$ |
| | AUC-PR | $0.595_{\pm 0.065}$ | $0.604_{\pm 0.004}$ | $0.581_{\pm 0.045}$ | $0.559_{\pm 0.016}$ |
| | AUC-ROC | $0.853_{\pm 0.021}$ | $0.877_{\pm 0.002}$ | $0.860_{\pm 0.019}$ | $0.824_{\pm 0.078}$ |

Table 7: Performance of Anomaly Detection on the Real-World MSDS Dataset

| Metrics | TranAD | USAD |
|---|---|---|
| F1 | $0.863_{\pm 0.114}$ | $0.888_{\pm 0.097}$ |
| AUC-PR | $0.995_{\pm 0.003}$ | $0.996_{\pm 0.001}$ |
| AUC-ROC | $0.981_{\pm 0.010}$ | $0.985_{\pm 0.003}$ |

We evaluate the performance of USAD for anomaly detection in terms of the F1 score, the area under the precision-recall curve (AUC-PR), and the area under the receiver operating characteristic (AUC-ROC) on two synthetic datasets. Table 6 shows the evaluation results on synthetic datasets, while Table 7 shows the evaluation results on the real-world MSDS dataset.

Overall, both USAD and TranAD can achieve promising performance on different types of anomalies and on both synthetic and real-world datasets, which lays a solid foundation for recourse prediction.

