# OpenReview forum: "Algorithmic Recourse in Abnormal Multivariate Time Series"
_TMLR — Accepted by TMLR_

### Review · Reviewer_mPHQ · 2025-09-03

**Summary Of Contributions:**

The paper proposes a new approach for algorithmic recourse, specifically for Time Series anomaly detection; which is yet unexplored and has practical application. The authors do a good job of motivating and describing the problem, and in  describing a clear framework.
The Granger Causality based approach is straightforward conceptually, and aligns with the general understanding of multivariate time series, along with GVAR that fits well into this framework. However, the assumptions are strong for applying to real world systems that may not be isolated, with additional factors.
The backtracking counterfactual argument is sensible in the problem formulation, as a the key motivation is often mitigation. The authors could better explain their take on structural interventions ( or at least clarify delegation to future work in a crisper manner).  The methodology is clearly presented, and technically sound. In the experimental section, however , the baselines seem not sufficiently meaningful. Also, it would be interesting to see how the system would work under presence of complex anomalous patterns such as Change Points, and distribution shifts. Moreover how does RecAD behave with different scoring mechanisms ( including paradigms such as Point Adjustment ) would be interesting to see as well. some of  metrics provided do not tie back to the traditional Algorithmic Recourse (or Algorithmic Recourse for Anomaly Detection). The counterfactuals generated here also do not have a diversity aspect, which is often important.
The paper's writing presents a clear narrative, however some doubts on the practical applicability remains.

**Audience:**

Yes

**Audience Explanation:**

Algorithmic Recourse is an active research area, and Algorithmic Recourse for anomaly detection is nascent. The work is novel in exploring a new territory.

**Claims And Evidence:**

Yes

**Claims Explanation:**

The manuscript clearly explains the technical contributions.

**Requested Changes:**

- Revise the section on metrics
- Address diversity of counterfactuals
- {optional) how different AD systems affect the performance of RecAD

---

> ### Author Response · Authors · 2025-10-07
>
> > RC1.
>
> We thank the reviewer for this comment. We have revised the Evaluation Metrics section to explicitly justify the choice of each metric. In particular: flipping ratio measures the effectiveness of the recourse by quantifying the proportion of anomalous time steps that are successfully reverted to normal after applying the intervention; action cost captures the feasibility of the recourse by measuring the overall magnitude of interventions required to flip anomalies; and action steps reflect the efficiency of the recourse by counting the number of discrete time steps at which interventions are applied.
>
> > RC2.
>
> Thank you for this comment. We acknowledge that diversity is an important consideration in algorithmic recourse, as it enhances robustness and user choice. While our current work does not explicitly promote diversity through traditional mechanisms, it can still achieve a degree of diversity by tuning the hyperparameter $\lambda$ in Eq. 7, which balances recourse effectiveness and action cost. Specifically, smaller values of $\lambda$ yield higher action costs but higher flipping ratios, whereas larger values produce lower costs and lower flipping ratios. By varying $\lambda$, the model generates multiple distinct recourse trajectories with different cost-effectiveness trade-offs, thereby introducing diversity among the counterfactuals, as illustrated in Fig. 5. We have added these discussions to the revised paper.
>
> > RC3.
>
> Thanks for this comment. We include a new anomaly detection model, TranAD, to check the performance of RecAD. As shown in our updated submission, our approach, RecAD, still outperforms baselines in various evaluation metrics by adopting TranAD as the score function for anomaly detection.
>
> > Comment: The authors could better explain their take on structural interventions ( or at least clarify delegation to future work in a crisper manner).
>
> Thanks for this suggestion. We have added a "Limitations and Future Work" section to discuss directions for handling structural interventions. In particular, we propose developing methods to learn a modified causal structure that, if implemented, would restore the system to normal behavior. This will involve designing an encoder–decoder architecture capable of adapting to changes in local causal mechanisms.

---

### Review · Reviewer_SqwT · 2025-09-16

**Summary Of Contributions:**

In order to address the lack of methods for algorithmic recourse in multivariate time series, RecAD proposed by the authors achieves effective and cost-efficient reversal of anomalies through framing recourse as a constrained maximum likelihood problem based on backtracking counterfactuals, but its effectiveness is contingent upon a pre-computed and accurate causal graph and is primarily designed for anomalies arising from external rather than structural interventions.

**Additional Comments:**

Dependency on Causal Discovery: The entire framework's performance is predicated on having an accurate Granger causal graph, which is learned beforehand using GVAR. If the GVAR model fails to capture the true causal dynamics, the recourse actions generated by RecAD could be suboptimal or incorrect. The paper does not analyze the model's sensitivity to errors in the causal graph.

**Audience:**

Yes

**Audience Explanation:**

Novel Problem Formulation: The paper pioneers the application of algorithmic recourse to multivariate time series, a domain with significant practical importance. The formulation using backtracking counterfactuals is theoretically sound and well-motivated for explaining anomalies as external shocks. Methodological Rigor: The end-to-end framework is elegant, combining a sophisticated recourse function with a causally-grounded procedure (AAP) for modeling downstream effects. This moves beyond naive prediction-based approaches. Comprehensive Evaluation: The experiments are thorough, using a mix of synthetic data for controlled analysis (point vs. sequence, external vs. structural anomalies) and real-world data for demonstrating practical relevance. The inclusion of ablation and sensitivity analyses further strengthens the empirical claims.

It fills a research gap in multivariate time-series anomaly retracing. It has high practical relevance in fields like healthcare, finance, and industry. The methodology is innovative, combining causal reasoning with counterfactual inference in an end-to-end framework.

**Broader Impact Concerns:**

Limited Handling of Structural Interventions: The paper's primary assumption is that anomalies are caused by external interventions. While a method for handling structural interventions is proposed by reframing them as an additive anomaly term (Section 4.5.2, page 7), the authors themselves state that its significance "is worthy of future study" (page 7). This remains a key limitation for anomalies caused by fundamental system changes (e.g., component failure). Assumption of Stationarity: The use of Granger causality models like GVAR typically assumes the underlying causal relationships are stationary over time. The framework may not perform well in systems with non-stationary or dynamically changing causal structures.

**Claims And Evidence:**

Yes

**Claims Explanation:**

Clear methodology: Well-defined problem formulation, the Abduction-Action-Prediction (AAP) process, and counterfactual posterior derivation, ensuring transparency. Causal reasoning integration: Treating anomalies as external interventions and using counterfactual reasoning enhances interpretability and effectiveness. Component validation: Ablation studies confirm the necessity of key components like the retracing function with temporal context and anomaly dynamics. Comprehensive experiments: Results on synthetic and real datasets consistently show RecAD outperforming baselines across multiple anomaly types and metrics. Case studies and sensitivity analysis: Demonstrations in ecological and distributed systems plus hyperparameter studies improve practical understanding. While limitations remain in causal graph dependency and structural intervention handling, the evidence convincingly supports the claims.

**Requested Changes:**

Strengthen the robustness of causal relationship modeling, particularly by providing a more in-depth discussion or potential improvements regarding the dependency on the predefined causal graph. Clarify and unify the problem formulation for sequence-level attribution actions to ensure better consistency with the algorithm description. Enhance the treatment mechanism for structural interventions by offering more concrete solutions for such anomalies or outlining a more comprehensive future research outlook. Expand the experimental analysis on baseline comparisons and cost functions to improve the completeness and persuasiveness of the experimental results.

---

> ### Author Response · Authors · 2025-10-07
>
> > RC1: Strengthen the robustness of causal relationship modeling, particularly by providing a more in-depth discussion or potential improvements regarding the dependency on the predefined causal graph.
>
> Thanks for the comments. We have added a "Limitations and Future Work" section to address these potential issues. Specifically, for the robustness of causal relationship modeling, we plan to integrate recent causal discovery advances to complement the causal relationship learned by GVAR, including invariance-based or multi-environment selection to reduce spurious edges, procedures that address latent confounders and nonstationarity, and hybrid models that combine constraint-based, score-based, and neural approaches.
>
> > RC2: Clarify and unify the problem formulation for sequence-level attribution actions to ensure better consistency with the algorithm description.
>
> We thank the reviewer for pointing out the need for greater clarity in our problem formulation. In the paper, we first introduce our framework in the context of point anomalies and then extend the discussion to sequence anomalies. This ordering was chosen deliberately for the sake of simplicity: point anomalies provide a straightforward setting to illustrate the intuition and mechanics of our attribution framework, before addressing the more complex case of sequence-level anomalies. Algorithm 1 in the Appendix applies to both cases. We have revised Sec. 4.5.1 to make this progression clearer.
>
> > RC3: Enhance the treatment mechanism for structural interventions by offering more concrete solutions for such anomalies or outlining a more comprehensive future research outlook.
>
> We thank the reviewer for this valuable suggestion. We agree that our current work leaves room for further exploration in structural interventions. We have included discussions about future work in this direction in the newly added "Limitations and Future Work" section. In particular, we propose developing methods to learn a modified causal structure that, if implemented, would restore the system to normal behavior. This will involve designing an encoder–decoder architecture capable of adapting to changes in local causal mechanisms.
>
> > RC4: Expand the experimental analysis on baseline comparisons and cost functions to improve the completeness and persuasiveness of the experimental results.
>
> Thank you for the helpful suggestion. We have incorporated an additional anomaly detection baseline, the transformer-based model TranAD, and repeated the experiments. The results have been added to the manuscript, which remain consistent with our original findings.

---

### Review · Reviewer_P2a8 · 2025-09-26

**Summary Of Contributions:**

The paper addresses algorithmic recourse in the context of time series data, i.e., identifying minimal changes that would alter the output of a predictive model. Unlike prior work that mainly focused on static data (e.g., images, tabular records), the authors emphasize that time series carry inherent temporal correlations: an intervention at time $t$ not only affects predictions at that time but may also propagate into the future. This motivates the need for specialized methods rather than direct adaptation from other modalities.

# Strengths

- The focus on time series recourse is well-motivated and timely: temporal propagation makes interventions fundamentally different from static data domains.
- The paper is generally well-written, structured, and easy to follow, with relevant references to prior work.
- Experimental validation demonstrates both feasibility and practical relevance.

# Weaknesses

- Notation and figures are sometimes confusing or inconsistent (see below). This may reduce accessibility to readers.
- The sensitivity analysis (Sec. 5.5) is somewhat unclear (see below).

**Audience:**

Yes

**Audience Explanation:**

The paper tackles an important problem and clearly formulates how this problem is under-addressed in the current litterature.

**Broader Impact Concerns:**

--

**Claims And Evidence:**

Yes

**Claims Explanation:**

Experiments are convincing, even if I do not understand why the sensitivity analysis is performed on synthetic data alone (see below).

**Requested Changes:**

# Critical

## Improve notation and figures:

* Fig 2b is misleading: if I understood correctly, the $\star$ notation is used for the counterfactual variables, whereas it seems in Fig 2b that $u$ and $u^\star$ just correspond to different timestamps. Maybe the figure could be refactored such that on one side of the axis we have the factual variables and on the other side the counterfactual ones (this is just a suggestion but at least the Figure should be made clearer)
* Also, the notation is not consistent: in Fig 2b and Sec 4.1, $W_t$ is the window that ends in $x_t$ and $V_t$, $u^\star$ etc start with $x_t$ so there is an overlap between them whereas in Sec 4.2 the past window is denoted $W_{t-1}$ from which $\theta_t$ is computed (and you stated above that you wanted to optimize $P(u^\star_t|W_t)$)
* In Eq. 7, do you actually optimize on $\|\theta_t\|_2$ or do you rather use $\|\theta_t\|_2^2$ that seems better suited for optimization purposes (especially when using gradient based optimization). Note that in (Von Kügelgen et al., 2023) they suggest to use squared Mahalanobis distance.

# Non-critical (but recommended)

## Clarify the sensitivity analysis in Sec. 5.5:

I don't understand the sensitivity analysis presented in Section 5.5: here, the authors study the impact of hyperparameter $\lambda$ on both the flipping ratio and the "Action Value". If I understand well, the "action value" is the same as the action cost defined in Sec. 5.1. If so, it seems very difficult to reach an interesting trade-off between flipping ratio and action cost since both metrics seem to be highly correlated. Why isn't the same experiment presented on the real-world MSDS dataset?

## Improve presentation:

* Putting the GVAR loss function inline makes it difficult to read, please make some space for an actual isolated equation
* USAD acronym is used (Sec 5.1) before being properly introduced (Sec 5.2)

---

> ### Author Response · Authors · 2025-10-07
>
> > Critical 1.
>
> Thank you for capturing this, and your understanding is correct. To address it, we have replotted Fig. 2(b) so that $u^*$ and $u$ now include subscripts to indicate their range. Note that the factual exogenous variables $u_t, \ldots, u_{t+M}$, which should colocate with $u^\*$, remain omitted from the figure for simplicity.
>
> > Critical 2.
>
> Thank you for capturing this. In fact, this is not an inconsistency. In Section 4.1, we derived our objective, as shown in Eq. (3), which contains the term $P(u_t^* \mid W_t)$ with $u_t^* = u_t + \theta_t$. This objective is then implemented in Section 4.2 by incorporating both the **preceding temporal context** $W_{t-1}$ and the **current deviation information** $\Delta_t = x_t - \hat{x}_t$ to derive $\theta_t$. In short, $u_t^*$ remains a function of $x_t$ and depends on the **entire window $W_t$**.
>
> > Critical 3.
>
> Thanks for this comment. Yes, we are using $|\theta|_2^2$, the squared distance for optimization. We have updated the manuscript.
>
> > Non-critical 1.
>
> Yes, "Action Value" refers to the action cost. We have updated Fig. 5 and Sec. 5.5 accordingly for consistency.
>
> The trade-off between flipping ratio and action cost implies that the goal is to maximize the flipping ratio while minimizing the cost. These two quantities are naturally correlated: larger interventions typically yield higher flipping ratios. Our sensitivity analysis illustrates this relationship. In our method, the balance is controlled by the parameter $\lambda$. We can observe that beyond a certain point, increasing the cost produces only marginal improvements in the flipping ratio. This helps practitioners choose $\lambda$ values near the turning point of the curve to achieve the best balance between effectiveness and cost.
>
> On the other hand, in relation to Reviewer mPHQ's comment, this trade-off also provides a means of introducing diversity in counterfactuals. By varying $\lambda$, the model generates multiple distinct recourse trajectories with different cost-effectiveness trade-offs, which may also enhance user choice. We have added the related discussions to the updated manuscript.
>
> We used synthetic datasets for the sensitivity analysis because they provide ground-truth labels after interventions, enabling a rigorous evaluation of the trade-off between flipping ratio and action cost.

---

### Author Response · Authors · 2025-10-07

We sincerely thank all reviewers for their thoughtful comments. A detailed, point-by-point response to each reviewer's comments is provided below, and the revised manuscript has been uploaded for your consideration.

---

### Decision · Action_Editor_Q1oo · 2026-01-05

**Recommendation:** Accept as is

**Audience:**

Yes

**Audience Explanation:**

Algorithmic recourse is an active research area and its adaptation to anomaly detection is underexplored. Hence, the approach proposed in this work is of interest to the community and worth sharing with the community given the methodological contributions and the convincing experimental validation.

**Claims And Evidence:**

Yes

**Claims Explanation:**

This work considers algorithmic recourse in the context of time series data, i.e., identifying minimal changes that would alter the output of a predictive model. Due to the temporal nature of time series data there is a need for specialised methods. All reviewers indicated that this work is novel and well-grounded. Its integration of causal reasoning, counterfactual inference, and temporal anomaly modeling represents a meaningful conceptual advance beyond static recourse approaches.

Reviewers further highlighted the clear methodology, the novel problem formulation, and the solid technical contributions. The authors addressed all the concerns raised by the reviewers in their response and revised manuscript. The only concern that remained is the practical applicability of the method -- this was only mentioned by one of the three reviewers.